# Central Nervous System Involvement in Systemic Autoimmune Rheumatic Diseases—Diagnosis and Treatment

**DOI:** 10.3390/ph17081044

**Published:** 2024-08-07

**Authors:** Aline Santana Juncker, Simone Appenzeller, Jean Marcos de Souza

**Affiliations:** 1Department of Internal Medicine, School of Medical Sciences, Universidade Estadual de Campinas (UNICAMP), Campinas 13083-881, Brazil; a271960@g.unicamp.br; 2Post-Graduate Program in Medical Sciences, School of Medical Sciences, Universidade Estadual de Campinas (UNICAMP), Campinas 13000-000, Brazil; 3Department of Orthopedics, Rheumatology and Traumatology, School of Medical Sciences, Universidade Estadual de Campinas (UNICAMP), Campinas 13083-887, Brazil; appenzel@unicamp.br

**Keywords:** rheumatic diseases, connective tissue diseases, systemic vasculitis, central nervous system

## Abstract

Central nervous system (CNS) involvement in autoimmune rheumatic diseases represents a significant challenge for clinicians across all specialties. While most reviews on the subject focus on neurological manifestations within a specific rheumatic disease, few descriptions shift from neurological clinical syndromes to achieve rheumatological diagnoses. This narrative review aims to synthesize current knowledge on the diagnosis and management of CNS manifestations occurring in the most prevalent rheumatic conditions in adults. We searched the MEDLINE database using the terms “central nervous system”, “rheumatic diseases”, “systemic lupus erythematosus”, “rheumatoid arthritis”, “Sjögren syndrome”, and “vasculitis”. The search strategy included review articles from 2019 to 2024, published in English, Spanish, or Portuguese. We explored the pathophysiological mechanisms linking autoimmunity to CNS pathology, emphasizing the role of syndromic reasoning, autoantibody profiles, and imaging modalities as tools for diagnosis and determination of inflammatory activity. The review also discusses differential diagnoses through a stepwise approach to neurological syndromes, summarized in diagnostic flowcharts, and presents updated treatment options. Although our approach is primarily semiology-based, the complexity of the subject invites future endeavors involving new technologies, such as functional MRI, MR spectroscopy, and nuclear medicine.

## 1. Introduction

For clinicians, central nervous system (CNS) manifestations of systemic diseases are always a subject of concern. These manifestations are polymorphic and difficult to isolate, and the potential for permanent damage to vital structures adds urgency. Various pathogenic mechanisms can be responsible for CNS lesions or dysfunction, such as cerebrovascular disease, mass effect, biochemical toxicity, electrical disorders, or inflammation. Inflammation can be largely restricted to the CNS in primary CNS autoimmune diseases [1], but rheumatic conditions contribute significantly to CNS inflammatory conditions through various mechanisms [2].

### 1.1. Vasculopathy

Broadly speaking, vasculopathy is responsible for a significant portion of CNS manifestations in rheumatic diseases [3]. Vascular occlusion of the arterial bed can occur due to inflammation directed at the vessel wall, as seen in primary vasculitides [4], or due to immune complex deposition [3]. The mechanisms leading to vessel wall damage can be mediated by immune cells and their inducing cytokines, such as interleukin 6 (IL-6) in large-vessel vasculitis [5] or tumoral necrosis factor (TNF) in Behçet disease (BD) [6,7], or by complement activation and/or antibody-mediated toxicity, as seen in systemic lupus erythematosus (SLE) and antineutrophil cytoplasm antibody (ANCA)-associated vasculitis (AAV) [5,8]. When inflammation occurs inside the vessel, thrombosis is also facilitated, especially in the presence of antiphospholipid antibodies (aPL) [8]. Occlusion of the venous bed is also possible and is well documented in BD and antiphospholipid syndrome (APS) [5,9]. Vascular occlusion can also result from atherosclerotic mechanisms, which are further augmented by inflammation and oxidative stress. The clinical syndrome presented by the patient is related to the area and size of the vascular involvement, as will be discussed further.

### 1.2. Blood–Brain Barrier Dysfunction

The CNS is somewhat separated from the immune system by the blood–brain barrier (BBB). Therefore, extensive infiltration of the parenchyma by inflammatory cells is unusual [10]. Nonetheless, circulating autoantibodies or immune complexes can facilitate the expression of adhesion molecules and the migration of cells into the CNS [11]. Additionally, microglial cells and astrocytes can be sources of cytokines and chemokines responsible for BBB disruption [10,11]. Neuroinflammation caused by these events can lead to subtle disruptions in neural networks without gross anatomical consequences [10]. However, in a subset of patients, it can lead to severe neuronal impairment and/or death, or demyelination, which hinders neuronal function and can ultimately result in death and/or scarring [12,13], causing irreversible damage. The structures related to neuroinflammation will correlate with the clinical picture and will be discussed in detail throughout the review.

### 1.3. Extra-Parenchymal Disease

The infiltration of structures intrinsically related to the CNS can lead to neurological manifestations such as cranial nerve palsy, myelopathy, or meningitis. The mechanisms leading to the inflammation of these structures generally follow the same principles, usually mediated by cytokines, chemokines, and antibodies. For instance, overexpression of transforming growth factor β (TGFβ) is associated with fibrosis of the dura mater [14] and compression of adjacent structures. Additionally, high levels of TNF can cause the erosion of the vertebral support and collapse of the medullary channel, leading to compressive myelopathy [2].

### 1.4. Other Mechanisms

Organ failure can occur after an immune attack on specific organs, such as the kidneys and liver. This advanced dysfunction can lead to electrolyte imbalances, acid–base disturbances, toxic neuropathy, and hypoperfusion due to the dysregulation of blood pressure. All these phenomena can manifest as CNS symptoms. Additionally, rheumatic patients are more prone to infections, neoplasms, and bleeding, either due to disease activity or the treatments used [4].

In adults with autoimmune rheumatic diseases, inflammatory CNS involvement is present in approximately 20% of patients with SLE [8], 6% with Sjögren’s disease (SD) [2], 10% with BD [15], and 24% with primary systemic vasculitides [16]. It is important to highlight that these prevalences refer only to processes of true inflammatory origin. When all CNS manifestations are considered, including psychiatric and functional disorders, these numbers are much higher, with studies reporting their occurrence in up to 95% of SLE patients [17]. Therefore, it is crucial for every clinician treating rheumatic patients to possess the necessary diagnostic skills to differentiate between neurological syndromes and their rheumatic etiologies.

Recently, our group published an article describing a diagnostic workup focused on the peripheral nervous system manifestations of rheumatic autoimmune diseases [18]. The aim of that article was to present an inversion of the typical clinical reasoning for rheumatic diseases. While many publications describe the possible neurological manifestations of each autoimmune rheumatic condition, it is rare to find the opposite approach: establishing a rheumatologic diagnosis based on neurological syndromes. We believe this approach can be useful for undiagnosed patients in neurology or internal medicine clinics and for guiding the monitoring of neurological activity in patients already diagnosed with rheumatic diseases. This work follows the same principles but focuses on CNS manifestations. For each type of CNS syndrome, we also describe treatment strategies.

## 2. Methodology

We searched the MEDLINE database using the terms (and their respective Medical Subject Headings (MeSH) terms): “central nervous system” AND “rheumatic diseases” OR “systemic lupus erythematosus” OR “rheumatoid arthritis” OR “Sjögren syndrome” OR “vasculitis.” The search was conducted on 8 May 2024, and included review articles from 2019 to 2024, published in English, Spanish, or Portuguese, focusing on adult patients. The relatively small timeframe was chosen arbitrarily to aim for updated evidence and because we estimated that the results would yield sufficient material since we were covering many different diseases. One of the authors screened the results by title, aiming to include only full texts and exclude duplicates and unrelated publications. The remaining articles were listed using the PubMed (https://pubmed.ncbi.nlm.nih.gov/, accessed on 8 May 2024) tool “best match,” which prioritizes the most relevant publications, in blocks of 50 articles, a quantity also arbitrarily chosen. The first block of abstracts was assessed by all the investigators to cover all the intended clinical conditions outlined above. If necessary, new blocks of 50 abstracts would be analyzed until enough information covering all the diseases was available. The researchers were instructed to focus on studies dedicated primarily to diagnosis and treatment and, to a lesser extent, pathogenesis. Because the strategy was narrowed to review articles, the data started to become redundant, and only one block of abstracts was necessary. Finally, the selected articles were fully assessed to form the main source of the review. If necessary, additional articles from the pool were individually analyzed. For each specific neurological syndrome, we described the most probable associated rheumatic conditions and provided a reasonable diagnostic flow. Furthermore, treatment options are outlined for each rheumatic-disease-associated CNS manifestation.

## 3. Results and Review

The search yielded 481 articles. The author JMS checked for duplicates and unrelated articles by title, resulting in 111 articles with retrievable full texts. Using the “best match” tool of the PubMed interface, abstracts of the first 50 articles were analyzed by all the researchers, and 24 articles were initially selected and fully assessed for the narrative review. If necessary, additional articles from the pool were individually analyzed.

### 3.1. Large-Vessel Cerebrovascular Disease (LVCVD)

These manifestations account for the disruption of blood flow through large to medium-sized arteries of the CNS, whether due to vascular thrombotic occlusion or true vasculitis. Because large territories are involved in the ischemic phenomenon, common clinical manifestations include sudden neurological deficits related to the zone of ischemia. It is important to highlight that cardiovascular risk is usually elevated in rheumatic patients [19], making it challenging to attribute the event to immune-mediated mechanisms instead of an atherosclerotic stroke. Clinical parameters that could help differentiate these situations include the occurrence of the event within two years from the onset of rheumatic disease [8,20] and a temporal relationship with other systemic features, such as nerve, skin, or kidney disease, suggesting systemic vasculitis [16].

When considering inflammatory mechanisms, SLE is the most important rheumatic condition to keep in mind. This is not only because it is relatively common but also because CNS manifestations are fairly frequent in SLE, as discussed above. Additionally, CVD is reported to occur in up to 20% of patients, and those with aPL are at higher risk [17,21]. Systemic vasculitides, specifically those affecting large and medium vessels, should also be considered. Polyarteritis nodosa (PAN), Takayasu arteritis (TAK), and giant cell arteritis (GCA) are all related to vasculitis and/or cerebral hypoperfusion [16,22].

The diagnosis of CNS vascular occlusion is relatively straightforward with magnetic resonance imaging (MRI). Sequences such as diffusion-weighted imaging (DWI) and T2-weighted with fluid-attenuation inversion recovery (T2-FLAIR) will show acute infarctions and changes in white matter, respectively [23]. If the clinical picture suggests SLE (e.g., a young woman with skin lesions or glomerulonephritis), tests for antinuclear antibodies (ANA) and aPL (including lupus anticoagulant, anticardiolipin, and anti-β-2-glycoprotein I) could be useful [24]. If large-vessel primary vasculitides are suspected (TAK and GCA), computed tomography angiography (CTA) and magnetic resonance angiography (MRA) of the main aortic branches can show vessel stenosis, occlusions, and vessel wall thickening and edema [23]. Labeled fluorodeoxyglucose positron emission tomography (FDG-PET) is also an alternative for detecting wall inflammation [25]. Temporal artery ultrasound (US) or biopsy might be necessary in selected patients to confirm GCA [26]. Finally, patients with PAN can undergo abdominal imaging and frequently have skin or peripheral nervous system disease, allowing for the histological analysis of those sites [27]. A suggested flowchart for diagnosis is presented in Figure 1.

### 3.2. Treatment of LVCVD

Broadly, the management of stroke in patients with rheumatic diseases follows the same principles as in the general population. Substantial effort should be made to select patients for thrombolytic therapy and to control classic cardiovascular risk factors (e.g., diabetes, hypertension, smoking, etc.). If the stroke occurred as an isolated event (i.e., without other symptoms or signs of systemic disease activity), it is presumed that immune-mediated events contributed less to the thrombus formation, so immunosuppression is probably not necessary. Therefore, patients can be managed with antiplatelet and lipid-lowering therapy [28]. However, it is important to highlight that, for patients with aPL positivity, anticoagulation with vitamin K antagonists is preferred over antiplatelet therapy [29]. Additionally, in antiphospholipid syndrome patients, anticoagulation with direct oral anticoagulants (DOACs) is usually not recommended, as current evidence suggests that it may predispose a significant portion of patients to an increased risk of new thrombotic events [29].

In patients with active TAK or GCA, stroke is more often related to carotid inflammatory disease than cerebral arteries, so a careful assessment of cervical vessels is warranted. If the disease is considered active, besides conventional treatment for stroke, these patients should receive high-dose glucocorticoids, followed by tapering [30]. TAK patients should also receive glucocorticoid-sparing drugs, such as methotrexate (MTX) [30]. In patients with presumptive inflammatory CVD related to SLE or PAN, remission induction is usually attempted with methylprednisolone pulses combined with cyclophosphamide (CYC) [17,27,31]. For SLE, maintenance therapy consists of hydroxychloroquine, oral, fast-tapered glucocorticoids, and mycophenolate mofetil (MMF) or azathioprine (AZA) [31]. For PAN, maintenance includes oral glucocorticoids, MTX, or AZA [27].

To facilitate, we present in Table 1 a suggestion of induction and maintenance, with doses and duration.

### 3.3. Small-Vessel Cerebrovascular Disease (SVCVD)—Acute Onset

These manifestations result from the rapid obstruction of many terminal arteries or capillaries by thrombi or vessel inflammation [32]. Because the parenchymal area involved in the ischemic process is microscopic, these occlusive phenomena infrequently produce abrupt neurological deficits. Most patients will present with seizures, movement disorders, acute psychosis, or acute confusional states [1,4]. However, small-vessel vasculopathy leading to an area of infarction and hemorrhage can cause a stroke-like syndrome, as reported in antineutrophil cytoplasm antibody (ANCA)-related vasculitis (AAV) [16] and SLE [32].

Contrary to large-vessel cerebrovascular disease, which can result from chronic inflammation or treatment side effects, acute SVCVD is closely related to disease onset and activity [8,16,33]. Thus, neurologists should promptly search for skin, kidney, eye, ear, or throat manifestations in these individuals.

Seizures are more frequently episodic and tonic–clonic [17,33]. Epilepsy develops in a minority of patients [17], likely related to residual brain damage. The diagnostic workup includes the following: (1) imaging to exclude mass-effect entities, posterior reversible encephalopathy syndrome (PRES), granulomatous lesions, or demyelinating disease; (2) electroencephalography (EEG) to document the electrical phenomena; and (3) cerebrospinal fluid (CSF) analysis to exclude infection (especially viral). Although MRI might reveal structural explanations for electrical instability, many cases will have normal or nonspecific findings [33]. This is likely due to the microscopic nature of the disease, which MRI may not effectively assess [8]. EEG will behave similarly, with less than half presenting typical epileptiform patterns [33].

Movement disorders and psychosis are uncommon in the context of small-vessel vasculitis and, when present, are more related to connective tissue diseases. The diagnosis is mainly clinical, with brain imaging and CSF analysis performed to exclude differentials [17]. Brain single-photon-emission computed tomography (SPECT) could identify perfusion deficits [33], but it is usually not necessary for diagnosis and management.

Acute confusional states might emerge from direct neurological activity but more commonly result from metabolic, hydrostatic, or infectious conditions, all of which are fairly common in patients with disseminated vasculitis. Therefore, a complete biochemical analysis is warranted, along with CSF analysis to assess for infection [33].

Regarding etiologies, acute-onset SVCVD is strongly related to SLE, especially in younger patients [1]. Primary vasculitides are also possible culprits, with AAV leading the list. Among these, granulomatosis with polyangiitis (GPA) is the most associated [34], though one retrospective study also reported a high prevalence in patients with eosinophilic granulomatosis with polyangiitis (EGPA) [35]. Therefore, serology testing should encompass these entities through a critical approach. For SLE, ANA testing is always warranted. Additionally, aPL are well documented in patients with neuropsychiatric conditions [8]. Anti-ribosomal P protein has limited sensitivity [36] but is related to seizures and psychiatric conditions [8,33], and anti-endothelial cell antibodies could specifically relate to psychosis [8]. Anti-Smith antibodies, highly specific for SLE, have been correlated with severe manifestations, including seizures and psychosis [37]. Finally, regarding AAV, antiproteinase 3 antibody is more common than, though still possible, anti-myeloperoxidase.

### 3.4. Treatment of Acute-Onset SVCVD

The treatment of acute-onset SVCVD suggested here focuses on evidence from SLE, which is the predominant underlying condition in most patients. Many patients with SLE and SVCVD are severely ill, often with diffuse and active disease. For instance, if there is kidney involvement, treatment for this manifestation will likely also cover neurological disease. The mainstay therapy for life-threatening neuropsychiatric systemic lupus erythematosus (NPSLE) involves inducing remission with methylprednisolone pulses combined with CYC or rituximab (RTX) [17,31]. Maintenance therapy includes hydroxychloroquine (as in all SLE patients), oral fast-tapered glucocorticoids, and MMF or AZA [31]. Biological drugs such as anifrolumab and belimumab, currently used for skin and kidney disease, are not yet recommended for NPSLE [31]. As most NPSLE patients are positive for aPL, consideration should be given to low-dose aspirin or warfarin in these cases [29,31].

Briefly, for AAV, induction therapy is similar with methylprednisolone and CYC or RTX [38]. The small-molecule avacopan, a C5a receptor antagonist, is available for remission induction in patients with GPA or MPA [38], although its benefits in neurologic involvement are uncertain. For maintenance therapy, RTX is generally preferred [38]. Specifically for EGPA, the anti-interleukin 5 mepolizumab is an option based on limited evidence [39]. For suggested dosages, refer to Table 1.

### 3.5. Progressive SVCVD

Within this group are manifestations possibly related to slow and progressive microvascular occlusion [8], although other mechanisms are also suggested, particularly for SLE, such as autoantibody toxicity, dysfunction of the blood–brain barrier, microglial activation, excitatory amino acid toxicity, and oxidative stress [8,11]. Clinically, these manifestations present as mood and anxiety disorders and cognitive dysfunction. Often, these manifestations have a multifactorial substrate with limited immune mediation, but 3–5% of SLE patients develop severe cognitive dysfunction [17]. To facilitate the objective diagnosis of cognitive impairment, the American College of Rheumatology (ACR) has proposed a neuropsychological battery for diagnosing cognitive dysfunction in SLE [40]. However, assessment is complex, and definitions and grading of impairment remain heterogeneous across studies over recent decades [41]. Specific guidelines for patients with AAV were not found. In clinical practice, except for very severe cases, the diagnosis of these syndromes is usually clinical and often not associated with disease activity.

### 3.6. Treatment of Progressive SVCVD

There are no specific recommendations for managing mood disorders or cognitive impairment in SLE [17,20], and guidelines specifically addressing these issues in AAV patients were not found. However, strict control of common cardiovascular risk factors is essential, and patients with aPL may benefit from low-dose aspirin for primary prophylaxis against major events [42]. Regarding dementia, a controlled trial found no benefit of memantine in SLE [43]. Lastly, mood disorders should be managed according to standard psychiatric recommendations.

### 3.7. Optic Neuropathy

This morbid manifestation can be provoked either by ischemic optic nerve neuropathy [5,17] or optic nerve inflammation and demyelination [44]. A less common mechanism, occurring in about one-third of cases related to rheumatic conditions, is pachymeningitis [45,46], which will be discussed further (see Section 3.11). In vascular forms, onset is typically abrupt, unilateral, usually painless, with potential for consecutive bilateral involvement. *Amaurosis fugax* has also been reported [47]. In cases of optic nerve inflammation (optic neuritis), vision impairment can develop more insidiously over hours to days, often accompanied by eye pain and dyschromatopsia. While unilateral optic neuritis is common, simultaneous bilateral neuritis has also been documented [48]. Physical examination may reveal visual field loss and, frequently, an afferent pupillary defect in unilateral cases. Funduscopic examination may show optic nerve swelling in intraocular disease but may appear normal in cases of retrobulbar involvement [49], which can occur in both inflammatory and vascular mechanisms.

Diagnosis of optic neuropathy is primarily clinical, based on history and ophthalmologic examination, usually sufficient to distinguish optic nerve disease from other ocular and orbital entities. Gadolinium-enhanced cranial MRI is likely to show contrast enhancement in cases of optic neuritis [50], which would be absent in vascular neuropathy. The same MRI can also aid in detecting other CNS lesions, common in rheumatic diseases.

In terms of etiology among rheumatic diseases, optic neuropathy is a significant feature of GCA, also included in classification criteria [26]. Among systemic connective tissue diseases, SLE and SD are most commonly associated with optic neuropathy [51,52], occurring in 1% and 16% of cases, respectively [17,53]. While eye involvement is prevalent in GPA, typically causing painful manifestations such as scleral redness, proptosis, orbital inflammation, or extrinsic motor abnormalities [4], these are easily differentiated from optic nerve disease. In EGPA, a group reporting high brain manifestation prevalence also noted a very high incidence of ischemic optic neuropathy, around 54% [35]. Therefore, our suggested approach encompasses these entities.

For GCA, ultrasound of the temporal artery is recommended. According to recent ACR/European Alliance of Associations for Rheumatology (EULAR) guidelines, a hypoechoic, thickened superficial temporal artery wall (halo sign) is a reliable surrogate for artery biopsy [26]. Additionally, CTA or MRA of head, neck, and thoracic vessels can reveal wall thickening, stenoses, and occlusions [23]. Serum biomarkers of inflammation such as C-reactive protein (CRP) and erythrocyte sedimentation rate (ESR) are typically elevated. For SLE and SD, helpful antibodies include ANA, aPL, anti-ribosomal P protein, anti-Ro60/SSA, and anti-La/SSB [8,17,48]. Special consideration should also be given to anti-aquaporin 4 (AQP4) and anti-myelin oligodendrocyte glycoprotein (MOG) antibodies, as they may occur in a subset of SLE patients and could relate to demyelinating forms (see Section 3.9). Lastly, if suspicion of a rheumatic condition persists, ANCA testing may be beneficial. A diagnostic decision tree is proposed in Figure 2.

### 3.8. Treatment of Optic Neuropathy

The involvement of the optic nerve is often considered a critical event in autoimmune conditions due to the risk of vision loss. Therefore, initial treatment typically involves pulsed intravenous methylprednisolone followed by gradually tapering doses of oral prednisone, regardless of the underlying cause [17,48,54]. However, the choice of glucocorticoid-sparing agents may vary slightly depending on the specific condition.

For GCA, tocilizumab shows the strongest evidence, although methotrexate is widely used based on retrospective studies [54,55]. In SLE, SD, and EGPA, CYC or RTX may be added to pulsed methylprednisolone during the induction phase [17,31,48].

For maintenance therapy in SLE, hydroxychloroquine with AZA or MMF is recommended [31]. In EGPA, while evidence is limited, options include RTX, MTX, AZA, or mepolizumab [39]. In SD, recent EULAR guidelines emphasize the importance of glucocorticoid-sparing agents without specifying a preferred drug [56]. If a patient has received RTX for the induction of remission and still requires prednisone within six months, based on clinical experience, consideration should be given to another cycle of RTX for maintenance therapy.

Table 1 summarizes dosage and duration recommendations.

### 3.9. Inflammatory Myelopathy

The catastrophic and disabling nature of this condition within autoimmune rheumatic diseases is believed to stem from a combination of direct neuronal damage caused by inflammation and vasculopathy [20,57]. The term “transverse myelitis” is commonly used in this context [58], although not all lesions are strictly transverse anatomically, and inflammation may not always be confined to the medulla itself but can also affect its vessels, occasionally leading to misinterpretations. Typically, onset is acute, and the clinical presentation varies widely depending on the site of the lesion, encompassing symptoms such as paraparesis, quadriparesis, sphincter dysfunction, and sensory deficits [52].

Importantly, optic neuritis associated with myelitis has been reported as secondary to rheumatic disorders like SD and SLE [17,48].

The diagnostic workup to confirm myelopathy includes spinal imaging, typically gadolinium-enhanced MRI, to rule out compressive etiologies [17]. Infectious etiologies should also be considered, with viral and spirochetal origins potentially requiring CSF analysis [59]. To focus on rheumatologic causes, SD should be considered first [48], followed by SLE [17,20], and less frequently, Behçet’s disease (BD) [15] and AAV [34]. A similar diagnostic approach to optic neuropathy applies here, including testing for ANA, aPL, anti-ribosomal P protein, anti-Ro60/SSA, anti-La/SSB, anti-aquaporin 4 (AQP4), anti-myelin oligodendrocyte glycoprotein (MOG), and ANCA.

Diagnosis of BD, however, remains clinical, with a positive pathergy test or concurrent brainstem inflammation potentially reinforcing the diagnosis [13]. The diagnostic algorithm is summarized in Figure 2.

### 3.10. Treatment of Inflammatory Myelopathy

The confined dimensions of the spinal cord, which concentrate fibers from multiple CNS sites, contribute to the rapid debilitation of this condition. Apart from BD, initial treatment for other etiologies among rheumatic diseases typically involves pulsed methylprednisolone and CYC. RTX, intravenous immunoglobulin (IVIg), and plasmapheresis are also recommended as rescue therapies, although evidence supporting their efficacy is heterogeneous [17,20,31,56].

For BD, evidence favors AZA over CYC [60]; thus, the induction regimen may include methylprednisolone pulses and AZA. Additionally, anti-TNF antibodies are effective in neuro-BD [60], though they are not typically recommended for SLE, SD, or AAV.

Maintenance therapy principles align with those used for managing other life-threatening manifestations. Specifically, hydroxychloroquine is recommended for SLE patients whenever possible, while myelopathy may benefit from glucocorticoid-sparing agents such as MMF or AZA [31]. For patients with antiphospholipid syndrome, antiaggregation or anticoagulation therapy should be considered based on bleeding risk [17].

For SD, maintenance therapy recommendations are lacking [56], but experts often extrapolate treatment strategies from those used for SLE. In AAV, achieving remission typically involves RTX as the preferred agent [38]. Similarly, for BD, AZA is commonly used in both induction and maintenance phases [15,60].

Specific dosages and treatment schemes are detailed in Table 1.

### 3.11. Meningeal Disease

This section discusses aseptic meningitis (AM) and hypertrophic pachymeningitis (HP). AM is characterized by a triad of headache, fever, and meningeal irritation, evidenced by clinical signs or CSF pleocytosis [17,20]. It results from inflammation within the inner meningeal compartment, including the arachnoid space, mediated by immune responses or vasculitis [61].

HP is a rare condition with an unclear pathogenesis, characterized by the progressive fibrotic thickening of the dura mater [14]. Autoimmune markers such as B-cell activating factor (BAFF), an A proliferation-inducing ligand (APRIL), and transforming growth factor beta (TGF-β) may be involved [62]. External compression of the dura can also lead to venous congestion, arterial compression, and focal scarring [14], often presenting clinically with insidious headache and neurological deficits, frequently involving cranial nerves, particularly the optic nerve [14].

The diagnosis of AM involves CSF analysis, typically showing mild pleocytosis, elevated proteins, normal glucose levels, and negative cultures and molecular tests for viral (e.g., herpes virus, enterovirus), bacterial (especially *Mycobacterium* spp.), and fungal (especially *Cryptococcus* spp.) infections [63]. HP is usually diagnosed via MRI, revealing abnormally thickened dura with potential gadolinium enhancement in primary inflammatory conditions [14].

AM is associated with various rheumatic autoimmune conditions. Among systemic connective tissue diseases, rheumatoid arthritis (RA), SLE, and SD are frequently implicated [17,45,48]. BD and AAV are common among vasculitides [64]. Sarcoidosis is also noted among miscellaneous autoimmune disorders [64].

In RA, extra-articular manifestations, including meningeal involvement, are associated with seropositivity to rheumatoid factor (RF) and anti-citrullinated peptide antibodies (ACPA) [45]. Diagnostics for SLE, SD, and AAV include serological markers such as antinuclear antibodies (ANA), anti-DNA antibodies, anti-ribosomal P protein antibodies, anti-Ro60/SSA antibodies, anti-La/SSB antibodies [8,48], and ANCA. Diagnosis of BD may be aided by MRI due to frequent parenchymal manifestations.

HP, as a rare manifestation of autoimmune rheumatic diseases, requires consideration of pre-test probability and judicious investigation. A focal thickened dura may not fully accounted for nonspecific neurological symptoms. HP is described in AAV, IgG4-related disease (IgG4-RD), sarcoidosis, and RA [14,45,62]. AAV and RA feature serological biomarkers ANCA, RF, and ACPA. However, diagnosing IgG4-RD and sarcoidosis often necessitates histopathological evaluation, as dura mater involvement alone is insufficient. IgG4-RD criteria involve extensive exclusion, including negative results for ANCA, anti-SSA/Ro, anti-SSB/La, anti-DNA, anti-Sm antibodies, and cryoglobulinemia [65]. Elevated IgG4 serum levels support the diagnosis but are not definitive [65]. Consideration of salivary gland, pancreatic, and retroperitoneal involvement is crucial for IgG4-RD, with chest computed tomography aiding in sarcoidosis and IgG4-RD diagnostics [62,65]. Figure 3 outlines a suggested diagnostic approach.

### 3.12. Treatment of Meningeal Manifestations

AM poses a diagnostic challenge due to its similarity to infectious meningitis, necessitating careful evaluation, particularly in patients lacking typical features of autoimmune rheumatic diseases (e.g., skin, joint, kidney, peripheral nerve, or lung involvement), especially for insidious etiologies such as tuberculosis or cryptococcosis, particularly in immunosuppressed individuals. Specific guidelines for presumed autoimmune meningitis are currently lacking. In conditions like BD and AAV, AM represents a potentially severe condition, warranting aggressive therapy with pulsed methylprednisolone or high-dose prednisone (see Table 1) [15,38]. Similar aggressive treatment approaches may be considered for AM associated with other rheumatic conditions.

Glucocorticoid-sparing agents vary depending on the underlying disease. Hydroxychloroquine is essential in SLE, with EULAR guidelines recommending CYC or rituximab RTX for severe neuropsychiatric manifestations [31]. CYC or RTX are also recommended similarly for AAV and SD [38,56], while MTX is preferred for RA and sarcoidosis [66,67]. For BD, AZA and anti-TNF agents are recommended [60].

Hypertrophic pachymeningitis (HP) follows similar treatment principles, but due to its proliferative nature, antiproliferative agents such as CYC or RTX are preferred over glucocorticoids alone [38,68].

It is important to note that nonsteroidal anti-inflammatory drugs (NSAIDs) should be avoided in the management of meningeal diseases due to their potential to exacerbate meningitis [20]. Detailed treatment schemes and dosages are provided in Table 1.

**Table 1 pharmaceuticals-17-01044-t001:** Suggested schemes for induction and maintenance of remission of the discussed autoimmune rheumatic diseases.

Drug	Indication	Induction Phase	Duration	Maintenance Phase	Duration	Observation
**Low-dose aspirin**	APS, aPL positivity [29]	n/a	n/a	75–100 mg/day	undetermined	
**Warfarin**	APS [29]	n/a	n/a	Daily dose necessary for INR between 2 and 3	undetermined	Dose should be slowly titrated, according to INR
**Intravenous immunoglobulin**	IM, encephalitis [17,56]	1 g/Kg/day	2 days	n/a	n/a	
**Intravenous methylprednisolone**	LVCVD, acute-onset SVCVD, ON, IM, meningeal disease, encephalitis, and demyelinating disease [17,20,31,38,56]	250–1000 mg/day	3 days	n/a	n/a	
**Intravenous cyclophosphamide**	SLE and SD-related syndromes [31,56]	500 mg every 15 days	3 months	n/a	n/a	
	RA and BD-related syndromes [45,60]	500 mg every 15 days or 0.75–1 g/m^2^ every month	3 months or 6 months respectively	n/a	n/a	
	AAV-related syndromes [38]	15 mg/kg every 2 weeks for 3 doses followed by 15 mg/kg every 3 weeks for at least 3 doses	2.5–6 months	n/a	n/a	
**Prednisone**	All discussed conditions [17,20,31,38,45,56,60]	n/a	n/a	1 mg/kg/day up to 80 mg/day	undetermined	Aim for 5 mg/day in 19th week
**Methotrexate**	all RA cases, vasculitides [38,66]	n/a	n/a	10–25 mg/week	undetermined for RA, at least 18 months for AAV, up to 18 months for PAN	
**Mycophenolate mofetil**	SLE-related syndromes [31]	2–3 g/day	3–6 months	2–3 g/day	at least 18 months	
**Azathioprine**	Vasculitides, SLE, and SD-related syndromes [31,38,56]	n/a	n/a	2–3 mg/kg/day	undetermined	
**Rituximab**	AAV, RA, SD, SLE, and IgG4D [31,38,56,66,68]	1g on days 1 and 15	15 days (2 doses)	500 mg every 6 months	undetermined, at least 24 months for AAV	

Abbreviations: AAV: antineutrophil cytoplasm antibody (ANCA)-associated vasculitis; aPL: antiphospholipid antibodies; APS: antiphospholipid syndrome; BD: Behçet’s disease; IgG4D: IgG4-related disease; IM: inflammatory myelopathy; LVCVD: large-vessel cerebrovascular disease; ON: optic neuropathy; PAN: polyarteritis nodosa; RA: rheumatoid arthritis; SD: Sjögren’s disease; SLE: systemic lupus erythematosus; SVCVD: small-vessel cerebrovascular disease.

### 3.13. Demyelinating Disease and Encephalitis

Demyelinating diseases like multiple sclerosis (MS) and neuromyelitis optica (NMO) are characterized by the loss of myelin sheath with relatively preserved axons [69]. In SD, CNS involvement manifests with diverse presentations, including transverse myelitis, optic neuropathy, and MS-like lesions [48]. The presence of anti-Ro antibodies in MS patients suggests potential cross-reactivity with viral or myelin antigens [69]. CSF analysis can aid in distinguishing SD from MS, showing fewer oligoclonal bands in SD compared to the typical oligoclonal pattern in MS. Standard diagnostic tools for SD, such as the Schirmer test and salivary gland biopsy, are valuable for differentiation [44]. CNS involvement in BD can also present with features resembling MS [69]. The treatment of SD aims to relieve symptoms and control the disease with immunosuppressive medications, such as corticosteroids, conventional synthetic disease-modifying drugs, and biological agents, as discussed before. Besides medications, non-pharmacological care is important to improve patients’ quality of life [48].

Autoimmune encephalitis, a rare complication of autoimmune diseases [70], manifests with systemic symptoms and can involve headaches, neuropsychiatric alterations, and may progress to seizures and coma [45]. The treatment of autoimmune encephalitis aims to suppress the aberrant immune response and control inflammation. Initial therapeutic strategies include corticosteroids for acute inflammation reduction, supplemented by other immunosuppressants such as CYC, AZA, MMF, and RTX [45]. The specific treatment varies based on severity and specific conditions. In SLE, hydroxychloroquine is universally recommended, with severe CNS involvement necessitating initial intravenous methylprednisolone followed by oral prednisone. Pulsed CYC induction has demonstrated long-term efficacy [31]. AZA and MMF serve as maintenance options to reduce steroid use and prevent relapses, while RTX is considered for refractory cases and can be used for both induction and maintenance. IVIg or plasmapheresis are options for refractory cases [2].

Treatment strategies for CNS involvement in SD are based on limited evidence. Initial treatment typically involves high-dose intravenous methylprednisolone followed by oral prednisone, with a gradual tapering regimen over weeks to months. Plasma exchange combined with steroids may be effective, particularly in cases of extensive transverse myelitis (LETM), considering potential overlap with neuromyelitis optica spectrum disorder (NMOSD) [2]. The 2020 EULAR guidelines for SD recommend CYC and plasma exchange combined with RTX as second and third-line options for severe CNS involvement [2,56]. Symptomatic management with anticonvulsants, antidepressants, and neurological rehabilitation may also be necessary.

## 4. Conclusions

CNS disease related to autoimmune rheumatic conditions is multifactorial. During their development, its patients will experience both inflammatory and non-inflammatory phenomena, making the etiological reasoning very challenging even for the experienced clinician. Imaging modalities, such as MRI and PET-CT, play a crucial role on the determination of topography and extent of the involvement. In addition, serological markers are helpful and necessary, but frequently not enough to determine the disease or the inflammatory status of the event.

Considering these challenges, we presented an unusual framework to address them. However, we recognize that generalizing the clinical approach based on neurological syndromes limits our ability to delve into specific pathogenic phenomena, making the contribution of our work to the state of the art limited. Additionally, our research was limited in terms of considering the molecular and genetic features of the discussed entities. For instance, immune signature profiles and genetic polymorphisms are becoming more clinically available and will undoubtedly be part of the diagnostic arsenal early in the investigation. Finally, we only briefly described more recent functional imaging due to limited evidence. Future endeavors involving new technologies, such as functional MRI, MR spectroscopy, or nuclear medicine, will hopefully facilitate our task. Nevertheless, a critical approach to clinical reasoning and physical examination is still the best weapon to confront these intriguing entities.

## Figures and Tables

**Figure 1 pharmaceuticals-17-01044-f001:**
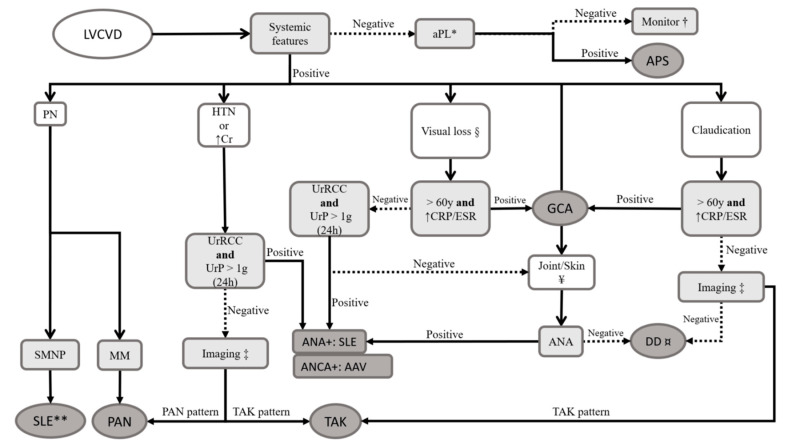
Proposed approach for the patient with large-vessel cerebrovascular disease. * Two positive results for lupus anticoagulant, anticardiolipin, or anti-β-2-glycoprotein I, at least 12 weeks apart [9]. † Treat for conventional stroke and monitor for systemic features. Consider primary neurological syndromes (i.e., primary angiitis of the central nervous system). § Attributable to optic neuropathy. ‡ Thoracoabdominal computed tomography or magnetic resonance angiography. ¥ Clinically inflammatory arthritis and erythematous lesions. ¤ Consider reassessing cardiovascular risk factors for ischemic disease. ** By classification criteria, antinuclear antibodies are required for systemic lupus erythematosus diagnosis [24]. If isolated LVCVD is considered (i.e., an abrupt event of clinically significant arterial thrombosis without other inflammatory systemic features), APS is a plausible diagnosis if aPL are present. Consider primary angiitis of the central nervous system if these antibodies are absent. If systemic features are present, the type of systemic involvement can suggest the etiology: (1) peripheral nerve disease leans towards PAN or, rarely, SLE; (2) glomerulonephritis suggests AAV or SLE; (3) optic neuropathy in elderly patients is highly suggestive of GCA, with SLE to be considered in younger patients, especially if skin or joint disease is associated; and (4) if limb or jaw claudication occurs, large-vessel vasculitis is probable, and GCA and TAK should be investigated. Always consider primary atherosclerotic stroke as well, given its higher prevalence compared to autoimmune conditions, even in younger patients. Abbreviations: AAV: antineutrophil cytoplasm antibody (ANCA)-associated vasculitis; ANA: antinuclear antibodies; aPL: antiphospholipid antibodies; APS: antiphospholipid syndrome; Cr: serum creatinine; CRP: C reactive protein; DD: differential diagnosis; ESR: erythrocyte sedimentation rate; GCA: giant cell arteritis; HTN: arterial hypertension; IgG4D: IgG4-related disease; LVCVD: large-vessel cerebrovascular disease; MM: multiple mononeuritis; PAN: polyarteritis nodosa; PN: peripheral neuropathy; SLE: systemic lupus erythematosus; SMNP: sensorimotor polyneuropathy; TAK: Takayasu arteritis; UrP: urinary protein; UrRCC: urinary red cell casts; y: years of age.

**Figure 2 pharmaceuticals-17-01044-f002:**
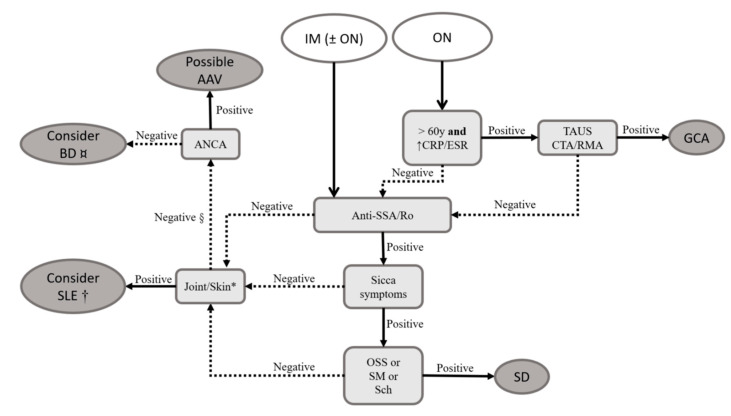
Proposed approach for the patient with optic neuropathy and/or inflammatory myelopathy. * Clinically inflammatory arthritis and erythematous lesions. † According to classification criteria, the diagnosis of systemic lupus erythematosus requires the presence of antinuclear antibodies [24]. Consider testing for anti-DNA, anti-Sm, anti-ribosomal P protein, and antiphospholipid antibodies. § The probability of rheumatic conditions is already low, especially if gadolinium-enhanced magnetic resonance shows optic nerve enhancement. Consider ordering anti-aquaporin 4 and anti-myelin oligodendrocyte glycoprotein for differential diagnosis. ¤ This applies specifically to cases of isolated inflammatory myelopathy. In the presence of ocular manifestations, consider consulting with an ophthalmologist, as Behçet’s disease rarely causes optic neuritis but is strongly associated with uveitis [15]. Epidemiologically, GCA is the most common systemic vasculitis in adults, and ON is a hallmark of the disease. Therefore, every elderly patient with acute-onset amaurosis should be evaluated for GCA. In younger patients, SLE and SD should be considered. Among cases of IM, including those with ON, GCA is unlikely, and SLE and SD are the main culprits. BD and AAV are alternative diagnoses, but it is important to note that BD rarely causes ON and more commonly leads to uveitis and retinal vasculitis, which can usually be differentiated through ophthalmologic examination. Abbreviations: AAV: antineutrophil cytoplasm antibody (ANCA)-related vasculitis; BD: Behçet’s disease; CRP: C reactive protein; CTA: computed tomography angiography; ESR: erythrocyte sedimentation rate; GCA: giant cell arteritis; IM: inflammatory myelopathy; ON: optic neuropathy; OSS: ocular staining score (ophthalmic exam with specific dyes with the aim to document and quantify eye surface’s loss of epithelialization); RMA: magnetic resonance angiography; Sch: Schirmer’s test (quantification of lacrimal flow volume by time unit); SD: Sjögren’s disease; SLE: systemic lupus erythematosus; SM: unstimulated sialometry (simple quantification of salivary flow volume by time unit); TAUS: temporal artery ultrasound; y: years of age.

**Figure 3 pharmaceuticals-17-01044-f003:**
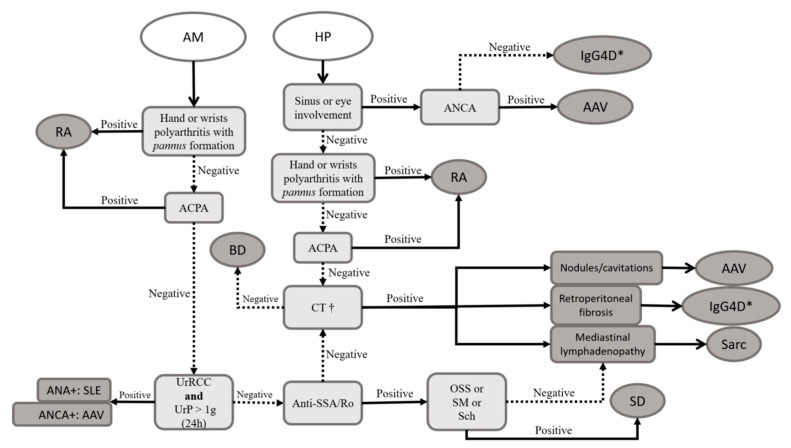
Proposed approach for the patient with aseptic meningitis or hypertrophic pachymeningitis. * The American College of Rheumatology (ACR)/European Alliance of Associations for Rheumatology (EULAR) classification criteria include an extensive exclusion criterion, requiring negativity for ANCA, anti-SSA/Ro, anti-SSB/La, anti-DNA, anti-Sm antibodies, and cryoglobulinemia [65]. † We suggest performing contrast-enhanced computed tomography of the thorax (for aseptic meningitis) or the thoracoabdominal region (for hypertrophic pachymeningitis). Because RA is the most prevalent autoimmune rheumatic disease worldwide and AM is well described in this condition, RA should always be considered as a potential final diagnosis. However, SLE, BD, sarcoidosis, and AAV are also possible diagnoses, making it challenging to narrow down AM among rheumatic diseases. Signs of glomerulonephritis and erythematous skin lesions can suggest SLE, while sinus involvement is more indicative of AAV. The eyes should be carefully examined by an ophthalmologist, as uveitis suggests BD, keratoconjunctivitis suggests SD, and scleritis or orbital involvement suggests AAV. Sarcoidosis typically affects the lungs and lymph nodes, which can guide diagnosis. When HP is found, RA, AAV, sarcoidosis, and IgG4D should be considered as potential causes. Pulmonary and mediastinal imaging can help narrow down the diagnoses, as typical patterns may emerge. Abbreviations: AAV: antineutrophil cytoplasm antibody (ANCA)-related vasculitis; ACPA: anti-citrullinated peptide antibodies; AM: aseptic meningitis; ANA: antinuclear antibodies; BD: Behçet’s disease; CT: computed tomography; HP: hypertrophic pachymeningitis; IgG4D: IgG4-related disease; OSS: ocular staining score (ophthalmic exam with specific dyes with the aim to document and quantify the eye surface’s loss of epithelialization); RA: rheumatoid arthritis; Sarc: sarcoidosis; Sch: Schirmer’s test (quantification of lacrimal flow volume by time unit); SD: Sjögren’s disease; SLE: systemic lupus erythematosus; SM: unstimulated sialometry (simple quantification of salivary flow volume by time unit); UrP: urinary protein; UrRCC: urinary red cell casts.

## Data Availability

Not applicable.

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
