# Peer review of "Central Nervous System Involvement in Systemic Autoimmune Rheumatic Diseases—Diagnosis and Treatment"

_pharmaceuticals, 2024, doi:10.3390/ph17081044_

Round 1

Reviewer 1 Report

Comments and Suggestions for Authors

The present review tries to show the implications of Central Nervous System Involvement in Systemic Autoimmune  Rheumatic Diseases, targeting the Diagnosis and Treatment. Numerous issues have been identified with the manuscript received, both in form and in substance, which make the present form of the manuscript unsuitable for publication.

1. First, it is very short for a comprehensive narrative review updating the state of knowledge in the field. 

2. The background in the abstract is poorly presented to mark the necessity of this review.

3. It is not clear why only the last 5 years were chosen for evaluation.

4. The concluding part of the abstract should be improved in terms of results and future research directions to which this research can refer.

5. The bibliographic structure and style required for this journal is not respected. It is recommended to re-read the instructions for authors and apply them in the template provided by the journal.

6. Chapters 2 and 3 are much too small and not useful presented in this form. It is necessary to present a single chapter on methodology, but better outlined and explained.

7. Table 1 is difficult to read and understand.

8. What is the meaning of the title of chapter 5?

9. Overall the supposed contribution to the literature is extremely small in this form of presentation. It is very important to emphasize novelty by updating the state of the art in the field, based on concrete data from in vitro and in vivo studies, as well as an extensive presentation of future research directions.

Author Response

The present review tries to show the implications of Central Nervous System Involvement in Systemic Autoimmune  Rheumatic Diseases, targeting the Diagnosis and Treatment. Numerous issues have been identified with the manuscript received, both in form and in substance, which make the present form of the manuscript unsuitable for publication.

  1. First, it is very short for a comprehensive narrative review updating the state of knowledge in the field.

We understand your concern. After all the adjustments, our manuscript now has 5,871 words. We believe it is now closer to the ideal length, considering that many respected medical journals set the word count for reviews at 6,000 words. Nevertheless, please consider that we did not aim to update the state of knowledge in the field, but rather to present a different point of view on the diagnosis and treatment of rheumatic diseases, departing from the neurological clinical syndromes related to them.

  1. The background in the abstract is poorly presented to mark the necessity of this review.

Thank you, we agree. The main objective of the manuscript is to present a different approach to the neurological manifestations of autoimmune rheumatic diseases. When searching scientific sources, one can easily find information on the neurological manifestations within each rheumatic condition (e.g., neurological manifestations of systemic lupus erythematosus). However, when facing a patient with a specific neurological syndrome but without a final diagnosis, clinicians can have trouble narrowing down the possible rheumatological culprits. Our main objective was to present this different framework for clinical reasoning. The abstract background now reads:

“Central nervous system (CNS) involvement in autoimmune rheumatic diseases represents a significant challenge for clinicians across all specialties. While most reviews on the subject focus on neurological manifestations within a specific rheumatic disease, few descriptions shift from neurological clinical syndromes to achieve rheumatological diagnoses.”

  1. It is not clear why only the last 5 years were chosen for evaluation.

The specific value was chosen preemptively, but it was arbitrary. Considering that we are covering broad features of many different diseases and that current knowledge evolves rapidly, we believed this strategy would cover the most important aspects, maintain an updated approach, and yet yield a reasonable number of articles given the narrative nature of the project. Indeed, 481 articles were retrieved. We have now highlighted this in the methodology section:

“The relatively small timeframe was chosen arbitrarily to aim for updated evidence and because we estimated that the results would yield sufficient material since we are covering many different diseases.”

  1. The concluding part of the abstract should be improved in terms of results and future research directions to which this research can refer.

We tried to cover that in the new concluding section of our Abstract, which now reads:

“The review also discusses differential diagnoses through a stepwise approach to neurological syndromes, summarized in diagnostic flowcharts, and presents updated treatment options. Although our approach is primarily semiology-based, the complexity of the subject invites future endeavors involving new technologies, such as functional MRI, MR spectroscopy, and nuclear medicine.”

  1. The bibliographic structure and style required for this journal is not respected. It is recommended to re-read the instructions for authors and apply them in the template provided by the journal.

We understand your concern. However, in the “Instructions for Authors” section of Pharmaceuticals’ homepage, the “Free format submission” subsection states: “Your references may be in any style, provided that you use consistent formatting throughout.” Thus, we’ve chosen AMA style. We will contact the Editorial Office to check if we should make amendments regarding this issue. Nevertheless, thank you for the advice. 

  1. Chapters 2 and 3 are much too small and not useful presented in this form. It is necessary to present a single chapter on methodology, but better outlined and explained.

We agree. In fact, this was a subheading issue after formatting. The 'Results and Review' (Section 3) is the longest section of the manuscript, with all neurological syndromes as subtopics. We have included a more detailed search strategy and corrected the subheadings. The new 'Methodology' section is provided below, and we invite you to review the entire manuscript formatting to understand the subheading corrections.

“We searched the MEDLINE database using the terms (and their respective Medical Subject Headings (MeSH) terms): “central nervous system” AND “rheumatic diseases” OR “systemic lupus erythematosus” OR “rheumatoid arthritis” OR “Sjögren syndrome” OR “vasculitis.” The search was conducted on May 8, 2024, and included review articles from 2019 to 2024, published in English, Spanish, or Portuguese, focusing on adult patients. The relatively small timeframe was chosen arbitrarily to aim for updated evidence and because we estimated that the results would yield sufficient material since we are covering many different diseases. One of the authors screened the results by title, aiming to include only full texts and exclude duplicates and unrelated publications. The remaining articles were listed using the PubMed tool “best match,” which prioritizes the most relevant publications, in blocks of 50 articles, a quantity also arbitrarily chosen. The first block of abstracts was assessed by all the investigators to cover all the intended clinical conditions outlined above. If necessary, new blocks of 50 abstracts would be analyzed until enough information covering all the diseases was available. The researchers were instructed to focus on studies dedicated primarily to diagnosis and treatment and, to a lesser extent, pathogenesis. Because the strategy was narrowed to review articles, the data started to become redundant, and only one block of abstracts was necessary. Finally, the selected articles were fully assessed to form the main source of the review. If necessary, additional articles from the pool were individually analyzed. For each specific neurological syndrome, we described the most probable associated rheumatic conditions and provided a reasonable diagnostic flow. Furthermore, treatment options are outlined for each rheumatic-disease-associated CNS manifestation.”

  1. Table 1 is difficult to read and understand.

We agree. We rotated the table to increase the text size. Additionally, we provided an editable version of the table to the Editorial Office, allowing the editorial team to adapt it according to the journal's layout.

  1. What is the meaning of the title of chapter 5?

Unfortunately, it was a formatting issue. The manuscript does not contain a Chapter 5.

  1. Overall the supposed contribution to the literature is extremely small in this form of presentation. It is very important to emphasize novelty by updating the state of the art in the field, based on concrete data from in vitro and in vivo studies, as well as an extensive presentation of future research directions.

We appreciate your honest and assertive opinion. Nevertheless, please consider that we did not aim to update the state of knowledge in the field, but rather to present a different point of view on the diagnosis and treatment of rheumatic diseases, departing from the neurological clinical syndromes related to them. Many articles, a large portion of which were cited in our work, have discussed in detail the state of the art of each entity on its own. Our manuscript is intended to be more of a practical guide for clinicians, synthesizing the information in a stepwise, clinically-friendly approach. We recognize, though, that our contribution to the scientific body is limited. We have included a limitations paragraph in the Conclusion section, as follows:

“Considering these challenges, we presented an unusual framework to address them. However, we recognize that generalizing the clinical approach based on neurological syndromes limits our ability to delve into specific pathogenic phenomena, making the contribution of our work to the state of the art limited. Additionally, our research was limited in terms of considering the molecular and genetic features of the discussed entities. For instance, immune signature profiles and genetic polymorphisms are becoming more clinically available and will undoubtedly be part of the diagnostic arsenal early in the investigation. Finally, we only briefly described more recent functional imaging due to limited evidence.”

We appreciate your contributions and the time you have dedicated to our manuscript. We hope that we have addressed your concerns and improved our manuscript accordingly.

Best regards.

Reviewer 2 Report

Comments and Suggestions for Authors

Authors conducted a comprehensive review to highlight the diagnostic and therapeutic approaches in autoimmune rheumatic diseases involving the central nervous system. The review in significant for the scientific community. However, authors are suggested to incorporate the changes to improve the quality of the article.

Comments:

1. Authors are suggested to provide the background and literature shortage on the current topic. The introduction should be improved by providing the sub headings and increase its length (no. of words). Authors should provide more balanced coverage of other conditions like rheumatoid arthritis, Sjögren's syndrome and vasculitis as well other than SLE.

2. Authors should ensure the consistent coverage of all mentioned diseases in the abstract and introduction.

3. Authors are suggested to include the explanations of how autoimmune processes lead to specific CNS manifestations and include the recent references.

4. The review should include recent advances in treatment options, such as biologics and targeted therapies, and their efficacy in CNS manifestations of rheumatic diseases.

5. Authors should ensure that all treatment recommendations must be via references. 6. Authors should add the limitations of the review article.

7. Ensure the format of the journal and consistent formatting throughout the article.

Author Response

Authors conducted a comprehensive review to highlight the diagnostic and therapeutic approaches in autoimmune rheumatic diseases involving the central nervous system. The review in significant for the scientific community. However, authors are suggested to incorporate the changes to improve the quality of the article.

Comments:

  1. Authors are suggested to provide the background and literature shortage on the current topic. The introduction should be improved by providing the sub headings and increase its length (no. of words). Authors should provide more balanced coverage of other conditions like rheumatoid arthritis, Sjögren's syndrome and vasculitis as well other than SLE.

We appreciate your valuable suggestions. Following your query #3, we decided to include a description of the pathogenic mechanisms in the introduction, which is now more comprehensive. Additionally, we provided a clearer explanation of our rationale for writing this manuscript, specifically addressing the presumed literature gap and focusing on CNS manifestations. Therefore, we kindly invite you to review our updated ‘Introduction’ section.

  1. Authors should ensure the consistent coverage of all mentioned diseases in the abstract and introduction.

Thank you for your highlight. Because our work is structured around neurological syndromes, each related rheumatic condition is distributed across various subsections. We are confident that all relevant conditions have been covered. However, it is important to note that some rheumatic diseases can cause multiple neurological syndromes, so they may appear frequently, while others will appear less often.

  1. Authors are suggested to include the explanations of how autoimmune processes lead to specific CNS manifestations and include the recent references.

Thank you. As mentioned in response to your query #1, this information is now included in the Introduction.

  1. The review should include recent advances in treatment options, such as biologics and targeted therapies, and their efficacy in CNS manifestations of rheumatic diseases.

We agree. Although the evidence is limited for many neurological syndromes due to their relative rarity, we discussed the role of biologics such as rituximab, anifrolumab, belimumab, tocilizumab, mepolizumab, and anti-TNF throughout the manuscript. Additionally, the targeted small molecule avacopan was briefly discussed in the 'Treatment of acute-onset SVCVD' section.

  1. Authors should ensure that all treatment recommendations must be via references.

We agree. All recommendations were made based on the most recent guidelines for each rheumatic condition. In Table 1, we have now added references to support our suggestions.

  1. Authors should add the limitations of the review article.

Thank you for the suggestion. We have included a limitations paragraph in the ‘Conclusion’ section, as follows:

“Considering these challenges, we presented an unusual framework to address them. However, we recognize that generalizing the clinical approach based on neurological syndromes limits our ability to delve into specific pathogenic phenomena, making the contribution of our work to the state of the art limited. Additionally, our research was limited in terms of considering the molecular and genetic features of the discussed entities. For instance, immune signature profiles and genetic polymorphisms are becoming more clinically available and will undoubtedly be part of the diagnostic arsenal early in the investigation. Finally, we only briefly described more recent functional imaging due to limited evidence.”

  1. Ensure the format of the journal and consistent formatting throughout the article.

We understand your concern. However, in the “Instructions for Authors” section of Pharmaceuticals’ homepage, the “Free format submission” subsection states: “Your references may be in any style, provided that you use consistent formatting throughout.” Therefore, we chose AMA style. We will contact the Editorial Office to check if amendments are needed. Nevertheless, thank you for the advice.

We truly appreciate your time and comprehensive suggestions.

Best regards.

Reviewer 3 Report

Comments and Suggestions for Authors

Review of the Manuscript: "Central Nervous System Involvement in Systemic Autoimmune Rheumatic Diseases – Diagnosis and Treatment"

The manuscript addresses a significant and complex issue in the field of rheumatology and neurology by exploring the involvement of the central nervous system (CNS) in systemic autoimmune rheumatic diseases. The authors aim to provide a comprehensive narrative review of the diagnostic and therapeutic strategies for these conditions. While the topic is highly relevant and the manuscript is well-structured, there are several areas that could benefit from improvement. The manuscript covers a crucial topic, providing a thorough review of CNS involvement in autoimmune rheumatic diseases, which is pertinent to both rheumatologists and neurologists.

My comments:

1. Methodology Section: Although the search terms and databases are specified, there is insufficient detail on the inclusion and exclusion criteria used during article selection. Providing more explicit criteria would enhance the reproducibility of the review.

The process of narrowing down from 481 articles to the final 24 is not thoroughly explained. More information on the reasons for excluding certain articles after the abstract review would improve transparency.

2. Results and Review:The manuscript mainly focuses on qualitative descriptions. Incorporating more quantitative data, such as prevalence rates of specific CNS manifestations in each rheumatic disease, would strengthen the findings.

The level of detail varies between sections. For instance, the discussion on large-vessel cerebrovascular disease is more comprehensive than other sections. Ensuring consistent detail across all conditions would provide a more balanced review.

3. Discussion: A more critical appraisal of the most recent studies and their impact on current understanding and practices would be beneficial. The manuscript could benefit from a more critical analysis of the reviewed literature, highlighting gaps, limitations, or controversies in current research.

4. Figures and Flowcharts: Some figures, such as diagnostic flowcharts, are difficult to read due to small font sizes and dense information. Simplifying these figures or providing more detailed legends could improve usability.

5. Conclusion: The conclusion section could better summarize the key diagnostic and therapeutic recommendations made throughout the manuscript. Discussing future research directions or unresolved questions in the field would provide a forward-looking perspective and add value to the manuscript.

Comments on the Quality of English Language

There are minor grammatical errors and formatting inconsistencies throughout the manuscript. A thorough proofreading and editing process is recommended.

Author Response

Review of the Manuscript: "Central Nervous System Involvement in Systemic Autoimmune Rheumatic Diseases – Diagnosis and Treatment"

The manuscript addresses a significant and complex issue in the field of rheumatology and neurology by exploring the involvement of the central nervous system (CNS) in systemic autoimmune rheumatic diseases. The authors aim to provide a comprehensive narrative review of the diagnostic and therapeutic strategies for these conditions. While the topic is highly relevant and the manuscript is well-structured, there are several areas that could benefit from improvement. The manuscript covers a crucial topic, providing a thorough review of CNS involvement in autoimmune rheumatic diseases, which is pertinent to both rheumatologists and neurologists.

My comments:

  1. Methodology Section: Although the search terms and databases are specified, there is insufficient detail on the inclusion and exclusion criteria used during article selection. Providing more explicit criteria would enhance the reproducibility of the review.

The process of narrowing down from 481 articles to the final 24 is not thoroughly explained. More information on the reasons for excluding certain articles after the abstract review would improve transparency.

You are absolutely correct. We have revised the ‘Methodology’ section, which now reads:

“We searched the MEDLINE database using the terms (and their respective Medical Subject Headings (MeSH) terms): “central nervous system” AND “rheumatic diseases” OR “systemic lupus erythematosus” OR “rheumatoid arthritis” OR “Sjögren syndrome” OR “vasculitis.” The search was conducted on May 8, 2024, and included review articles from 2019 to 2024, published in English, Spanish, or Portuguese, focusing on adult patients. The relatively small timeframe was chosen arbitrarily to aim for updated evidence and because we estimated that the results would yield sufficient material since we are covering many different diseases. One of the authors screened the results by title, aiming to include only full texts and exclude duplicates and unrelated publications. The remaining articles were listed using the PubMed tool “best match,” which prioritizes the most relevant publications, in blocks of 50 articles, a quantity also arbitrarily chosen. The first block of abstracts was assessed by all the investigators to cover all the intended clinical conditions outlined above. If necessary, new blocks of 50 abstracts would be analyzed until enough information covering all the diseases was available. The researchers were instructed to focus on studies dedicated primarily to diagnosis and treatment and, to a lesser extent, pathogenesis. Because the strategy was narrowed to review articles, the data started to become redundant, and only one block of abstracts was necessary. Finally, the selected articles were fully assessed to form the main source of the review. If necessary, additional articles from the pool were individually analyzed. For each specific neurological syndrome, we described the most probable associated rheumatic conditions and provided a reasonable diagnostic flow. Furthermore, treatment options are outlined for each rheumatic-disease-associated CNS manifestation.”

  1. Results and Review:The manuscript mainly focuses on qualitative descriptions. Incorporating more quantitative data, such as prevalence rates of specific CNS manifestations in each rheumatic disease, would strengthen the findings.

The level of detail varies between sections. For instance, the discussion on large-vessel cerebrovascular disease is more comprehensive than other sections. Ensuring consistent detail across all conditions would provide a more balanced review.

Thank you for your valuable suggestion. Where possible, we have included the prevalences of the specific neurological syndromes.

Regarding the size of each section, we encountered an issue with the formatting of our subheadings. We have made corrections to ensure the sections appear more homogeneous. We invite you to review the divisions throughout the manuscript to see if they are now more suitable.

  1. Discussion: A more critical appraisal of the most recent studies and their impact on current understanding and practices would be beneficial. The manuscript could benefit from a more critical analysis of the reviewed literature, highlighting gaps, limitations, or controversies in current research.

We agree. In response to your query #5, we have addressed this in the ‘Conclusion’ section by highlighting our strengths and limitations, as well as providing suggestions for future research.

  1. Figures and Flowcharts: Some figures, such as diagnostic flowcharts, are difficult to read due to small font sizes and dense information. Simplifying these figures or providing more detailed legends could improve usability.

Thank you for your attention to our figures. Modifying the flowcharts would be very exhaustive and would require us to reconsider our diagnostic approach. Therefore, we decided to explain the decision trees in detail. After each figure, a brief text is now provided to clarify the diagnostic flow.

  1. Conclusion: The conclusion section could better summarize the key diagnostic and therapeutic recommendations made throughout the manuscript. Discussing future research directions or unresolved questions in the field would provide a forward-looking perspective and add value to the manuscript.

Thank you. Please refer to your query #3 and review the updated sections ‘Introduction’ and ‘Conclusion.’

We truly appreciate your time and comprehensive suggestions.

Best regards.

Round 2

Reviewer 1 Report

Comments and Suggestions for Authors

The authors responded partially to some suggestions, not at all to others, trying to explain the lack of changes. It has been improved since the first revision, but for a journal with the pretensions and rigor of Pharmaceuticals, the changes made are not sufficient in my view.

Any review should bring an update of the state of knowledge in the field, to a greater or lesser extent depending on the topicality.

The explanation for the selection of the last 5 years leads to multiple biases.

The title of chapter 3 is confusing.

Regarding what you mentioned about bibliographic style it also says 'When your manuscript reaches the revision stage, you will be requested to format the manuscript according to the journal guidelines'. So it's more logical and simpler to do it right from the beginning, especially since when you download the template provided by the journal you can see exactly which style should be used.

Table 1 is still unclear.

I don't know if the most appropriate approach is the 'unusual framework to address them'.

Reviewer 2 Report

Comments and Suggestions for Authors

Suggestions have been incorporated. 

Reviewer 3 Report

Comments and Suggestions for Authors

The manuscript can be published in its current form.